

# Association between vitamin D receptor gene polymorphisms and athletic performance in Chinese male youth soccer players

Shidong Yang[1], Wei Zhang[1], Meng Jia[2] and Haichun Chen[3]

[1] Department of Physical Education, Nanjing Xiaozhuang University, Nanjing, China
[2] School of Physical Education and Sport Science, Nanjing Normal University, Nanjing, China
[3] School of Physical Education and Sport Science, Fujian Normal University, Fuzhou, China

## ABSTRACT

**Background.** The relationship between the vitamin D receptor (VDR) gene and muscle strength has been extensively investigated; however, the findings of this research remain inconclusive. The aim of this study was to evaluate the association between VDR variants (*ApaI* rs797523, BsmI rs1544410, and *FokI* rs2228570 genotypes) and athletic performance in youth soccer players in China.

**Materials and Methods.** A total of 142 male soccer players (73 from an elite group and 69 from a sub-lite group) aged 13–15 years, and 107 controls (13- to 14-year-old students) were recruited for this study. We measured height, weight, speed, explosive power, anaerobic endurance, and aerobic endurance in both the elite and sub-elite athletes. *Apa*I, *Bsm*I and *Fok*I genotypes were detected in controls, elite, and sub-elite soccer players with the single-nucleotide polymorphism (SNP) technique. The $\chi^2$ test was applied to analyze the correlation between genotype distribution and allelic frequency in elite and sub-elite athletes and controls. One-way analysis of variance and Bonferroni's post hoc test were implemented to analyze the differences in parameters among groups, and statistical significance was set at $p \leq 0.05$.

**Results.** (1) The genotype distributions of the *Apa*I, *Bsm*I, and *Fok*I in controls, elite, and sub-elite soccer players were consistent with the Hardy–Weinberg equilibrium (HWE) results, except for the *Bsm*I genotype distribution in control s ($\chi^2 = 7.396$, df = 1, $p = 0.025$). (2) The *Bsm*I AG frequency in the controls and sub-elite players was significantly higher than in the elite players ($\chi^2 = 6.4$, df = 1, $p = 0.011$; $\chi^2 = 4.50$, df = 1, $p = 0.034$, respectively). The frequency of the *Fok*I TT genotype in the controls was significantly higher than in the elite and sub-elite players ($\chi^2 = 12.737$, df = 1, $p < 0.001$, $\chi^2 = 8.805$, df = 1, $p = 0.003$, respectively). The frequency of the ApaI A in the elite players was significantly lower than that in the controls and sub-elite players ($\chi^2 = 3.765$, df = 1, $p = 0.05$; $\chi^2 = 12.19$, df = 1, $p < 0.001$ respectively). (3) *Apa*I CC players had longer distances in the standing long jump (SLJ) ($p = 0.026$) and shorter times in the 30-m run ($p = 0.003$) than *Apa*I AC players. Additionally, *Apa*I AA players had significantly longer Yo-Yo Intermittent Recovery Test Level 1 (YYIR1) running distances compared to *Apa*I AC players ($p = 0.002$).

**Conclusion.** Chinese elite youth soccer players are more likely to possess the *ApaI* CC genotype and are less likely than sub-elite players to have the *BsmI* A alleles.

Corresponding authors
Wei Zhang, weizhangtg@126.com
Haichun Chen, 1960105546@qq.com

Peer**J**

Additionally, the *Apa*I CC genotype is associated with better speed and explosive power among Chinese elite youth soccer players.

## INTRODUCTION

The vitamin D receptor (VDR) gene occupies an important position in the human genome. It is located in a specific region of chromosome 12 (12q12-q14) and is a relatively long gene segment, reaching a total length of 100 kb (*Taymans et al., 1999*). The VDR plays a key role in regulating the transcriptional activity of the vitamin D metabolite 1$\alpha$, 25-dihydroxyvitamin D3. Multiple polymorphisms can be found in the vitamin D receptor gene, including *Bsm*I, *Apa*I, and *Taq*I restriction sites at the 3′ end of the VDR gene, as well as a polymorphism initiation codon at the 5′ end of the VDR gene that can be identified using the restrictive endonuclease *Fok*I (*Habuchi et al., 2000*).

Vitamin D is essential for the regulation of calcium and phosphorus metabolism, and in maintaining bone health and skeletal muscle function. Vitamin D promotes muscle cell absorption of inorganic phosphates obtained from energy-rich phosphate compounds. These inorganic phosphates are vital to muscle contraction, and specific VDRs are located on the cell membrane and regulate and distribute intracellular calcium ions (*Houston et al., 2007*; *Windelinckx et al., 2007*). Studies have shown that moderate vitamin D supplementation exerts a positive impact on muscle function, facilitating adenosine triphosphate (ATP) regeneration, protein synthesis, and increased explosive power (*Wimalawansa, 2018*). The level of 25-hydroxyvitamin D3 (25-OH D3) in serum is directly related to human strength, speed, and explosive power (vertical jump) (*Larson-Meyer & Willis, 2010*; *Rejnmark, 2011*).

In adolescents, vitamin D is critical to the growth of bones and skeletal muscle, as well as neuromuscular function (*Vicente-Rodriguez et al., 2003*; *Li et al., 2020*). VDR gene polymorphisms affect the body's need for vitamin D to varying degrees. While vitamin D can be obtained through diet, it is primarily synthesized endogenously using the sun's ultraviolet B radiation. Vitamin D improves muscle mass and strength levels by increasing steroid hormone levels in the body *via* an enzymatic mechanism (*Willis, Peterson & Larson-Meyer, 2008*; *Yang et al., 2023*). An insufficient frequency and duration of light exposure can result in vitamin D deficiency in humans (*Grant & Soles, 2009*). However, it has been proposed that even adequate light frequency and duration may be insufficient for optimal vitamin D production (*Willis, Broughton & Larson-Meyer, 2009*).

Vitamin D deficiency has been linked to a number of adverse effects on muscle function, including muscle wasting, reduced muscle contraction, chronic muscle pain, and delayed recovery from muscle injury (*Bulgay et al., 2023*; *Shipton & Shipton, 2015*; *Pilch et al., 2020*). The administration of a moderate dose of vitamin D supplements has been demonstrated to mitigate the risk of muscle loss and bone fractures, while also enhancing muscle strength

in adolescents. The impact of vitamin D on adolescent athletes and professional soccer players is more pronounced during the winter season (*Perrone et al., 2024*; *Dubnov-Raz et al., 2014*). However, it is evident that vitamin D deficiency is not solely attributable to a lack of sunlight; genetic and epigenetic interactions also play a significant role in its pathogenesis.

Vitamin D and its receptor (Vitamin D Receptor, VDR) play a pivotal role in human physiology, particularly in the regulation of calcium and phosphorus metabolism, as well as in maintaining bone health and skeletal muscle function. In recent years, an increasing number of studies have revealed associations between vitamin D and athletic performance. For instance, vitamin D levels have been found to be directly correlated with indicators such as muscle strength, speed, and power (*Hamilton, 2011*; *Wiciński et al., 2019*). However, the relationship between VDR gene polymorphisms and athletic performance remains contentious. Although this issue has been explored in various populations, studies focusing specifically on adolescent soccer players are still limited. Studies have indicated that *ApaI* (*Flore et al., 2024*), *BsmI* (*Bozsodi et al., 2016*), and *FokI* (*Xia et al., 2019*) may be linked to muscle function. However, research focusing specifically on youth soccer players is still limited.

In light of this, the objective of this study was to evaluate the relationship between VDR gene variants (ApaI rs797523, BsmI rs1544410, and FokI rs2228570) and athletic performance in Chinese youth soccer players, and also to provide a scientific basis for the selection and training of adolescent soccer players.

## Research methods
### Subjects
A total of 167 players were initially recruited from two soccer schools and one professional soccer club in China. After excluding 15 goalkeepers and 10 players of non-Han ethnicity, a total of 142 players of Chinese Han ethnicity (62 13-year-olds, 47 14-year-old-olds, and 33 15-year-olds), were ultimately included. Additionally, we recruited 107 healthy, non-athletic junior high school boys aged 13–14 years as controls. Coaches defined players as starters or substitutes based on training and competitive ability. In this study, starters were defined as elite players and substitutes were defined as sub-elite players.

Data collection followed the procedures described in *Yang et al. (2023)*. Specifically, we conducted this study in accordance with the Declaration of Helsinki, and it was approved by the Ethics Committee of Fujian Normal University, with the approval number ''FNU-L 2024005''. All testing procedures were communicated to all players, their guardians, and coaches prior to testing and written informed consent was obtained.

## Experimental methods
This study was conducted over a two-day period. On the first day, from 08:00 to 10:00 AM, players' physical morphological indicators, such as height and weight, were measured. Players' buccal mucosa samples were also collected and preserved by team members Yang Shidong and Jia Meng. Initially, participants were asked to rinse their mouths with water before sample collection. Sterile flocked swabs were then used to gently scrape the oral mucosa for a minimum of 20 rotations to ensure adequate cellular collection. The

**Table 1  Primers used for *VDR ApaI, BsmI,* and *FokI* gene polymorphisms.**

|  | Primer F | Primer R |
|---|---|---|
| *VDR ApaI* (rs7975232) | ATCTGTGGGCACGGGGATAGA | TCTGGATCCTAAATGCACGGAGA |
| *VDR BsmI* (rs1544410) | CACTGCCCTTAGCTCTGCCTTG | CAGGAATGTTGAGCCCAGTTCA |
| *VDR FokI* (rs2228570) | TCTGGCTCTGACCGT | TTCCGGTCAAAGTCTCC |

swabs were then placed into sterile preservation tubes containing a preservative solution. Each sample was documented with the participant ID, collection time, location, and other relevant details. During transport to the laboratory, samples were kept in insulated containers at 2–8 °C. Upon arrival at the laboratory, the samples were promptly processed for genotypic analysis.

From 4:00 to 6:00 PM on the same day, participants completed a series of comprehensive tests assessing their physical performance. A standardized 15-minute warm-up and stretching session preceded the tests to prepare the athletes and reduce injury risk. The tests were conducted sequentially: a 30-meter sprint, a standing long jump (SLJ), a 5 × 25-meter shuttle run, and the Yo-Yo Intermittent Recovery Test Level 1 (YYIR1). A 5-minute rest period was provided between each test to allow for adequate recovery. Participants were instructed to warm up and stretch prior to the tests. All subjects were free of illness and injury and were able to complete the tests without any concerns. We extracted DNA from the subjects' buccal mucosa, and determined the genotypes of the three loci (*Apa*I rs797523, *Bsm*I rs1544410 and *Fok*I rs2228570) of the VDR by polymerase chain reaction (PCR).

## Testing procedures
### Genotyping
The oral mucosa was collected using a cased flocked swab (Longforce Bioinstruments, Shanghai, China) and placed in preservative solution. DNA was extracted from oral mucosa using a TSINGKE Silica Gel Adsorption Kit (Qiagen Inc., Valencia, CA, USA).

### PCR amplification
The primers were synthesized by Qingke Biotechnology Co., Ltd., and the sequences of the amplification primers are listed in Table 1. PCR amplification was performed using a 50-µL reaction system (Table 2). The PCR conditions and procedures for ApaI, BsmI, and FokI polymorphism analysis are depicted in Table 3.

The components of the amplification system were as follows: mix (green), 47 uL; 10 µM primer F, 1 uL; 10 µM primer R, 1 uL; and template (gDNA), 1 uL.

The amplification program consisted of an initial denaturation step at 98 °C for 2 min, followed by 30 cycles comprising 98 °C for 10 s, TM°C for 10 s, and 72 °C for 10 s, with a final extension step at 72 °C for 5 min. The amplified PCR products were then subjected to agarose gel electrophoresis (2 uL sample + 6 uL bromophenol blue) at 300 V for 12 min.

**Table 2  Components of the polymerase chain reaction (PCR) amplification system.**

| Components | Volume |
|---|---|
| Chingke Gold Mix | 47 μL |
| 10 μM Primer F | 1 μL |
| 10 μM Primer R | 1 μL |
| Template (gDNA) | 1 μL |
| Total | 50 μL |

**Table 3  PCR conditions for *VDR ApaI, BsmI,* and *FokI*.**

| Stage | Temperature | Time | Number of cycles |
|---|---|---|---|
| Pre-denaturation | 98 °C | 2 min | 1 cycle |
| | 98 °C | 10 s | |
| Cycle | TM °C | 10 s | 30 cycles |
| | 72 °C | 10 s | |
| Extension | 72 °C | 5 min | 1 cycle |
| Preservation | 4 °C | – | |

## Test protocol

Height, weight, and thigh circumference were measured to determine the study subjects' morphological body characteristics. The physical fitness of the subjects was evaluated by measuring their sprint, explosive, and aerobic endurance. Height and weight were measured once and thigh circumference was measured twice. Intraclass correlation coefficients (ICCs) were calculated according to *McGraw & Wong (1996)*. An ICC > 0.75 was defined as "almost perfect" (*McGraw & Wong, 1996*).

Data were collected as previously described in *Yang et al. (2023)*. Specifically, physical fitness of players was assessed in accordance with the Guidelines for Testing the Athletic Ability of Youth Football Players (Youth Training Syllabus, 2020), as set forth by the Chinese Football Association (CFA) (*Chinese Football Association, 2020*). This approach enabled an objective evaluation of the physical fitness requirements for Chinese youth soccer players, encompassing attributes such as speed, explosive power, and aerobic endurance. We tested the players with a 30-m sprint, SLJ, and 5 × 25-m repeat sprint ability (RSA), YYIR1, and 12-min running tests. All subjects were asked not to exceed their normal training load two days prior to the test to eliminate the effect of delayed muscle soreness on test performance.

All subjects completed testing in the following order: 30-m run, SLJ, 5 × 25-m RSA, and YYIR1, with the YYIR1 test performed last. Subjects' 12-min runs were tested at the same time the following afternoon. The 30-m sprint and the SLJ were tested twice, with a 3-min interval between tests. In contrast, the 5 × 25-m RSA, YYIR1, and the 12-min run were tested once. Before the formal test, players warmed up with 10 min of jogging and dynamic movements, followed by 5-min jumping and sprint running with progressive intensity.

## Statistical analysis

We conducted all statistical analyses using IBM SPSS 26.0 for Windows. The $\chi^2$ test was applied to examine the reports of Hardy–Weinberg equilibria (HWE) (*Shenoy et al., 2010*). At a significance level of $p \leq 0.05$, Fisher's exact tests were also implemented to examine genotypic distributions and allelic frequencies among control, elite, and sub-elite athletes. The significance of the observed differences was assessed using one-way analysis of variance (ANOVA), and Bonferroni *post hoc* tests were performed to determine which measurements showed significant differences. The difference in anthropometric and physical indices between elite and sub-elite athletes was assessed with Student's *t*-test.

## RESULTS

### Distribution of the ApaI, BsmI, and FokI genotypes in elite and sub-elite players

The genotypic distributions of elite players ($\chi^2 = 0.063$, $df = 1$, $p = 0.969$; $\chi^2 = 0.003$, $df = 1$, $p = 0.953$; and $\chi^2 = 4.38$, $df = 1$, $p = 0.112$, respectively), sub-elite players ($\chi^2 = 0.154$, $df = 1$, $p = 0.695$; $\chi^2 = 1.946$, $df = 1$, $p = 0.163$; and $\chi^2 = 1.468$, $df = 1$, $p = 0.226$, respectively), and controls ($\chi^2 = 0.002$, $df = 1$, $p = 0.999$; and $\chi^2 = 0.004$, $df = 1$, $p = 0.998$) were in agreement with the HWE, except for the BsmI genotypic distribution of controls ($\chi^2 = 7.396$, $df = 1$, $p = 0.025$).

Table 4 summarizes the distribution frequencies of *ApaI, BsmI,* and *FokI* genotypes and alleles in the control, elite, and sub-elite players; and we observed that the distribution frequencies of the *ApaI, BsmI,* and *FokI* genotypes in the groups were not significantly different. However, there was a significant difference in the distribution frequency of the *ApaI* AC and CC genotypes between the control, elite, and sub-elite players ($\chi^2 = 6.337$, $df = 2$, $p = 0.042$, $\chi^2 = 7.087$, $df = 2$, $p = 0.029$), with the controls and sub-elite players having significantly more AG frequencies ($\chi^2 = 6.4$, $df = 1$, $p = 0.011$; $\chi^2 = 4.50$, $df = 1$, $p = 0.034$, respectively). There was a significant difference in the distribution frequency of *FokI* TT genotypes among the controls, elite, and sub-elite players ($\chi^2 = 17.429$, $df = 2$, $p < 0.001$), with the frequency for TT genotypes in the controls significantly higher than that in the elite and sub-elite players ($\chi^2 = 12.737$, $df = 1$, $p < 0.001$, $\chi^2 = 8.805$, $df = 1$, $p = 0.003$, respectively); and there was a significant difference in the distribution frequency of *BsmI* GG genotypes ($\chi^2 = 8.393$, $df = 2$, $p = 0.015$).

There were also significant differences in the distribution frequency of *ApaI* and *BsmI* alleles among the controls, elite, and sub-elite players ($\chi^2 = 3.947$, $df = 1$, $p = 0.047$; $\chi^2 = 4.625$, $df = 1$, $p = 0.032$, respectively). The frequency of the ApaI A in the elite players was significantly lower than that in the controls and sub-elite players ($\chi^2 = 3.765$, $df = 1$, $p = 0.05$; $\chi^2 = 12.19$, $df = 1$, $p < 0.001$ respectively). The frequency of the *BsmI* A allele in the controls and sub-elite players were significantly higher than that in the elite players ($\chi^2 = 10.286$, $df = 1$, $p = 0.001$, $\chi^2 = 6.4$, $df = 1$, $p = 0.011$, respectively). The frequency of the *FokI* T allele in the controls was significantly higher than that in the elite and sub-elite players ($\chi^2 = 16.237$, $df = 1$, $p < 0.001$, $\chi^2 = 15.54$, $df = 1$, $p < 0.001$, respectively).

**Table 4 Frequencies and percentages of the distributions of *VDR ApaI* genotypes and alleles for controls, elite, and sub-elite players.**

| | Class | Control | Elite | Sub-elite | $\chi^2$ | df | P |
|---|---|---|---|---|---|---|---|
| | All (N) | 107 | 73 | 69 | | | |
| | AA (N/%) | 8/7.5 | 2/2.7 | 6/8.7 | 3.500 | 2 | 0.174 |
| | AC (N/%) | 42/39.3 | 22/30.1 | 31/44.9 | 6.337 | 2 | 0.042 |
| ApaI | CC (N/%) | 57/53.3 | 49/67.1 | 32/46.4 | 7.087 | 2 | 0.029 |
| (rs7975232) | HWE-*P* value | 0.998 | 0.969 | 0.695 | | | |
| | A allele (N/%) | 58/27.1 | 26/17.8 | 42/30.4 | 12.190 | 2 | 0.002 |
| | C allele (N/%) | 156/72.9 | 120/82.8 | 96/69.6 | 14.710 | 2 | 0.001 |
| | AA (N/%) | 2/1.9 | 0/0 | 1/1.4 | 0.333 | 2 | 0.564 |
| | AG (N/%) | 9/8.4 | 1/1.4 | 7/10.1 | 6.118 | 2 | 0.047 |
| BsmI | GG (N/%) | 96/89.7 | 72/98.6 | 61/88.4 | 8.393 | 2 | 0.015 |
| (rs1544410) | HWE-*P* value | 0.025 | 0.953 | 0.163 | | | |
| | A allele (N/%) | 13/6.1 | 1/0.7 | 9/6.5 | 9.739 | 2 | 0.008 |
| | G allele (N/%) | 201/93.9 | 145/99.3 | 129/93.5 | 18.055 | 2 | <0.001 |
| | CC (N/%) | 24/22.4 | 21/28.8 | 19/27.5 | 0.594 | 2 | 0.743 |
| | CT (N/%) | 53/49.5 | 44/60.3 | 39/56.5 | 2.221 | 2 | 0.329 |
| FokI | TT (N/%) | 30/28.0 | 8/11.0 | 11/15.9 | 17.429 | 2 | <0.001 |
| (rs2228570) | HWE-*P* value | 0.998 | 0.112 | 0.226 | | | |
| | C allele (N/%) | 101/47.2 | 86/58.9 | 77/55.8 | 3.341 | 2 | 0.188 |
| | T allele (N/%) | 113/52.8 | 60/41.1 | 61/44.2 | 23.564 | 2 | <0.001 |

## Odds ratios (ORs) for polymorphisms

Table 5 depicts the odds ratios (ORs) for VDR ApaI, BsmI, and FokI polymorphisms in controls, elite, and sub-elite players. The OR for elite players in the ApaI AA *vs.* CC genotype compared with sub-elite players was 0.218 (95% confidence interval (CI) [0.041–1.146]; $p = 0.05$); the OR for elite players in the ApaI CC *vs.*, (AA+AC) genotype compared with sub-elite players was 2.474 (95% CI [1.276–4.793]; $p = 0.007$); and the OR for ApaI CC *vs.* (AA+AC) genotype compared with elite players was 1.918 (95% CI [1.053–3.494]; $p = 0.03$).

The OR for elite players in the *BsmI* GG *vs.* (AA+AG) genotype compared with sub-elite players was 171.148 (95% CI [23.254–1,259.654]; $p < 0.001$); the OR for elite players in the *BsmI* GG *vs.* (AA+AG) genotype compared with controls was 9.75 (95% CI [1.247–76.252]; $p = 0.009$); and the OR for sub-elite players in the *BsmI* GG *vs.* (AA+AG) genotype compared with controls was 0.057 (95% CI [0.003–0.109]; $p < 0.001$).

The OR for elite players in the FokI CC *vs.* TT genotype compared with controls was 3.281 (95% CI [1.237–8.702]; $p = 0.015$); the OR for elite players in the FokI TT *vs.* (CC + CT) compared with controls was 0.313 (95% CI [0.313–0.719]; $p = 0.004$); the OR of sub-elite players in the FokI CC *vs.* (CC + TT) compared with controls was 0.238 (95% CI [0.16–0.352], $P = 0.015$). $p < 0.001$); and the OR for sub-elite players in the FokI TT *vs.* (CC + TT) compared with controls was 0.481 (95% CI [0.227–1.02]; $p = 0.05$).

**Table 5  Odds ratios of *VDR ApaI*, *BsmI* and *FokI* genotypes for control, elite, and sub-elite players.**

|  |  | Elite *vs.* Sub-elite | | Control *vs.* Elite | | Control *vs.* Sub-elite | |
|---|---|---|---|---|---|---|---|
|  |  | *P* | OR | *P* | OR | *P* | OR |
| *ApaI* | AA *vs.* AC | 0.37 | 0.470 (0.087; 2.548) | 0.37 | 0.477 (0.093; 2.443) | 0.98 | 1.016 (0.320; 3.228) |
|  | AA *vs.* CC | 0.05 | 0.218 (0.041; 1.146) | 0.11 | 0.291 (0.059; 1.434) | 0.62 | 1.336 (0.426; 4.192) |
|  | AA *vs.* (AC+CC) | 0.09 | 0.267 (0.053; 1.351) | 0.14 | 0.325 (0.068; 1.559) | 0.72 | 1.219 (0.410; 3.620) |
|  | CC *vs.* (AA+AC) | 0.007 | 2.474 (1.276; 4.793) | 0.03 | 1.918 (1.053; 3.494) | 0.40 | 0.775 (0.431; 1.395) |
| BsmI | AA *vs.* AG | 0.71 | 1.143 (0.880; 1.485) | 0.64 | 1.111 (0.904; 1.366) | 0.74 | 0.643 (0.048; 8.618) |
|  | AA *vs.* GG | 0.28 | 1.016 (0.985; 1.049) | 0.22 | 1.75 (1.535; 1.995) | 0.85 | 0.787 (0.070; 8.865) |
|  | AA *vs.* (AG+GG) | 0.07 | 1.008 (0.993; 1.023) | 0.77 | 1.045 (1.015; 1.075) | 0.84 | 0.779 (0.070; 8.679) |
|  | GG *vs.* (AA+AG) | <0.001 | 171.148 (23.254; 1,259.654) | 0.009 | 9.750 (1.247; 76.252) | <0.001 | 0.057 (0.03; 0.109) |
| FokI | CC *vs.* CT | 0.96 | 0.980 (0.460; 2.086) | 0.88 | 1.054 (0.519; 2.142) | 0.84 | 1.076 (0.518; 2.233) |
|  | CC *vs.* TT | 0.46 | 1.520 (0.505; 4.575) | 0.015 | 3.281 (1.237; 8.702) | 0.10 | 2.159 (0.864; 5.398) |
|  | CC *vs.* (CT+TT) | 0.75 | 1.124 (0.549; 2.298) | 0.14 | 1.648 (0.848; 3.201) | <0.001 | 0.238 (0.160; 0.352) |
|  | TT *vs.* (CC+CT) | 0.38 | 0.651 (0.249; 1.703) | 0.004 | 0.313 (0.136; 0.719) | 0.05 | 0.481 (0.227; 1.020) |

**Notes.**
Data are odds ratios (ORs) and 95% confidence intervals (CIs). Analysis was adjusted by competitive level.

## Association between genotype and physical fitness

As shown in Tables 6 and 7, the weight of players with the ApaI AA genotype was significantly higher than that for the AC genotype ($p = 0.04$), as was the case for sub-elite players. ApaI CC players' SLJ was significantly longer than that of AC players ($p = 0.023$), and. VapI CC players' 30-m running time was significantly shorter than that for AC players ($p = 0.009$).

ApaI AA players' YYIR1 running distances were longer than those for ApaI AC players ($p = 0.002$), and ApaI AA sub-elite players YYIR1 running distances were longer than for AC ($p = 0.008$) and CC players ($p = 0.03$). We discerned no difference in anthropometric or athletic performance in FokI CC, CT, or TT players between elite and sub-elite players.

## DISCUSSION

This study systematically investigated the association between VDR gene polymorphisms and athletic performance in Chinese youth soccer players. It was found that elite players were more likely to possess the *ApaI* CC genotype than sub-elite players, and that elite players were less likely than sub-elite players to have the *BsmI* AA and AG genotypes. The *ApaI* AA genotype was associated with weight and YYIR1 in our Chinese youth soccer players, and the ApaI CC genotype was associated with SLJ and 30-m sprint. The association of vitamin D with muscle activity and athletic performance has long been established (*Pfeifer, Begerow & Minne, 2002*). However, the association between vitamin D receptor gene polymorphisms and muscle strength remains controversial (*Bollen et al., 2023*).

Previous studies have extensively investigated the association between the VDR gene and muscle strength, particularly in the elderly population. *Krasniqi et al. (2025)* demonstrated that among elderly individuals in Kosovo, those with the *ApaI* CC genotype exhibited superior lower limb strength, as evaluated by the 30-second chair stand test and the

**Table 6  Comparison of anthropometrics and athletic performance among different *VDR ApaI* genotypes.**

| | Variables | VDR ApaI | | | F | p |
| | | AA | AC | CC | | |
|---|---|---|---|---|---|---|
| Overall | Height (cm) | 176.5 ± 7.0 | 168.8 ± 8.4 | 170.6 ± 8.9 | 2.877 | 0.06 |
| | Body mass (kg) | 61.6 ± 5.4[†] | 53.2 ± 9.2[†] | 55.1 ± 8.7 | 3.288 | 0.04 |
| | C thigh (cm) | 52.0 ± 3.7 | 50.1 ± 4.4 | 50.4 ± 5.1 | 0.568 | 0.568 |
| | VO$_2$ max (mL/kg/min) | 55.0 ± 4.9 | 54.4 ± 4.4 | 54.6 ± 4.4 | 0.082 | 0.921 |
| | 5 × 25-m RSA (s) | 34.2 ± 2.3 | 36.0 ± 2.7 | 35.1 ± 1.9 | 3.473 | 0.034 |
| | SLJ (cm) | 223.6 ± 22.7 | 214.8 ± 18.5[‡] | 223.8 ± 18.3[‡] | 3.779 | 0.025 |
| | 30 m (s) | 4.87 ± 0.30 | 4.79 ± 0.38[‡] | 4.59 ± 0.31[‡] | 6.623 | 0.002 |
| | YYIR1 (m) | 1,913.2 ± 268.2[†] | 1,654.5 ± 361.7[†‡] | 1,811.1 ± 306.0 | 6.957 | <0.001 |
| Elite | Height (cm) | 177.5 ± 4.9 | 171.4 ± 7.3 | 172.5 ± 8.8 | 0.524 | 0.595 |
| | Body mass (kg) | 66.1 ± 1.4 | 56.3 ± 8.8 | 57.4 ± 9.4 | 1.104 | 0.337 |
| | C thigh (cm) | 51.6 ± 2.2 | 51.7 ± 3.5 | 51.5 ± 4.7 | 0.007 | 0.993 |
| | VO$_2$ max (mL/kg/min) | 56.8 ± 5.2 | 55.7 ± 3.0 | 55.0 ± 5.2 | 0.273 | 0.762 |
| | 5 × 25-m RSA (s) | 33.6 ± 0.2 | 34.5 ± 2.0 | 34.4 ± 1.5 | 0.267 | 0.767 |
| | SLJ (cm) | 224.0 ± 14.1 | 226.8 ± 14.2 | 231.0 ± 15.0 | 0.766 | 0.469 |
| | 30 m (s) | 4.66 ± 0.06 | 4.59 ± 0.34 | 4.49 ± 0.24 | 1.206 | 0.355 |
| | YYIR1 (m) | 2,160.0 ± 0.0 | 1,708.2 ± 418.1 | 1,873.3 ± 293.6 | 2.834 | 0.066 |
| Sub-elite | Height (cm) | 176.2 ± 8.0 | 166.9 ± 8.8 | 167.7 ± 8.4 | 3.004 | 0.057 |
| | Body mass (kg) | 60.1 ± 5.5[†] | 50.9 ± 9.0[†] | 51.7 ± 7.0 | 3.431 | 0.038 |
| | C thigh (cm) | 52.2 ± 4.2 | 48.9 ± 4.7 | 48.7 ± 5.3 | 1.275 | 0.286 |
| | VO2 max (mL/kg/min) | 54.4 ± 5.1 | 53.5 ± 4.9 | 54.0 ± 3.0 | 0.219 | 0.804 |
| | 5 × 25 m RSA (s) | 34.5 ± 2.8 | 37.2 ± 2.6 | 36.2 ± 1.9 | 3.495 | 0.036 |
| | SLJ (cm) | 223.5 ± 26.0 | 206.1 ± 16.4 | 213.0 ± 17.6 | 2.818 | 0.067 |
| | 30 m (s) | 4.95 ± 0.31 | 4.93 ± 0.35 | 4.74 ± 0.34 | 2.661 | 0.077 |
| | YYIR1 (m) | 2,063.3 ± 187.4[†§] | 1,643.7 ± 307.3[†] | 1,710.6 ± 305.3[§] | 4.932 | 0.010 |

Notes.

VO$_2$ max, maximal oxygen uptake; SLJ, standing long jump; 5× 25-m RSA, 5× 25-m repeated sprint ability; YYIR1, YoYo intermittent recovery test level 1.

[†] Significant difference between the AA and AC genotypes; $p < 0.05$.

[‡] Significant difference between the CC and AC genotypes; $p < 0.05$.

[§] Significant difference between the AA and CC genotypes; $p < 0.05$.

6-minute walk test. *Wu et al. (2014)* concluded that subjects with the *ApaI* CC genotype reflected the greatest strength (grip), while those with the AC genotype exhibited the least strength, a finding consistent with the results of our study in elderly Taiwanese individuals. We ascertained that the players with the *ApaI* CC genotype manifested the best performance in speed and explosiveness, and that the players with the AC genotype demonstrated the worst performance with respect to speed and explosiveness. However, *Wang et al. (2006)* suggested that the *ApaI* AA genotype exhibited significantly lower knee and elbow concentric or eccentric peak torque than those with the aa or Aa genotype in Chinese female university students (*Wang et al., 2006*).

The influence of the ApaI polymorphism on human muscle strength remains inconclusive, likely due to differences in gender and ethnicity among subjects. According to *Ferrer-Suay et al. (2021)*, the ApaI site is located in the 3′ untranslated region of the eighth intron of the VDR gene. While this polymorphism does not alter the amino acid

**Table 7   Comparison of anthropometrics and athletic performance among different *VDR FokI* genotypes.**

| Groups | Variables | VDR FokI | | | F | p |
|---|---|---|---|---|---|---|
| | | CC | CT | TT | | |
| Overall | Height (cm) | 169.1 ± 9.6 | 170.9 ± 8.5 | 170.1 ± 8.1 | 0.571 | 0.566 |
| | Body mass (kg) | 53.9 ± 8.5 | 55.6 ± 9.5 | 53.3 ± 7.1 | 0.778 | 0.461 |
| | C thigh (cm) | 50.6 ± 4.6 | 50.6 ± 5.1 | 49.1 ± 3.6 | 0.723 | 0.487 |
| | VO$_2$max (mL/kg/min) | 54.1 ± 4.5 | 54.7 ± 4.6 | 54.7 ± 2.4 | 0.323 | 0.725 |
| | 5 × 25-m RSA (s) | 34.8 ± 2.5 | 35.7 ± 2.2 | 35.3 ± 2.1 | 2.205 | 0.114 |
| | SLJ (cm) | 222.2 ± 19.6 | 219.4 ± 18.9 | 221.9 ± 19.0 | 0.357 | 0.700 |
| | 30 m (s) | 4.64 ± 0.36 | 4.70 ± 0.37 | 4.66 ± 0.26 | 0.432 | 0.650 |
| | YYIR1 (m) | 1,779.5 ± 252.6 | 1,780.9 ± 363.5 | 1,724.4 ± 367.5 | 0.218 | 0.805 |
| Elite | Height (cm) | 171.7 ± 8.2 | 173.0 ± 8.1 | 169.7 ± 10.0 | 0.622 | 0.540 |
| | Body mass (kg) | 56.6 ± 8.0 | 58.4 ± 9.5 | 53.0 ± 7.9 | 1.365 | 0.262 |
| | C thigh (cm) | 52.5 ± 3.4 | 51.5 ± 4.5 | 49.5 ± 4.8 | 1.319 | 0.274 |
| | VO$_2$max (mL/kg/min) | 55.3 ± 3.0 | 55.3 ± 5.3 | 54.8 ± 2.8 | 0.024 | 0.977 |
| | 5 × 25-m RSA (s) | 33.9 ± 1.6 | 34.7 ± 1.6 | 34.2 ± 1.7 | 1.756 | 0.180 |
| | SLJ (cm) | 231.3 ± 13.8 | 228.8 ± 15.4 | 229.0 ± 14.8 | 0.206 | 0.814 |
| | 30 m (s) | 4.47 ± 0.24 | 4.55 ± 0.30 | 4.53 ± 0.23 | 0.675 | 0.513 |
| | YYIR1 (m) | 1,870.5 ± 259.7 | 1,813.2 ± 367.8 | 1,822.9 ± 436.7 | 0.196 | 0.823 |
| Sub-elite | Height (cm) | 166.2 ± 10.4 | 168.4 ± 8.4 | 170.4 ± 7.0 | 0.861 | 0.428 |
| | Body mass (kg) | 51.0 ± 8.2 | 52.3 ± 8.6 | 53.5 ± 6.9 | 0.356 | 0.702 |
| | C thigh (cm) | 48.5 ± 4.9 | 49.5 ± 5.5 | 48.8 ± 2.8 | 0.283 | 0.755 |
| | VO$_2$max (mL/kg/min) | 52.8 ± 5.4 | 54.1 ± 3.7 | 54.7 ± 2.5 | 0.986 | 0.379 |
| | 5 × 25-m RSA (s) | 35.9 ± 2.9 | 36.9 ± 2.2 | 36.1 ± 2.1 | 1.317 | 0.275 |
| | SLJ (cm) | 212.1 ± 20.3 | 208.7 ± 16.7 | 216.7 ± 20.7 | 0.883 | 0.418 |
| | 30 m (s) | 4.84 ± 0.38 | 4.87 ± 0.37 | 4.75 ± 0.24 | 0.514 | 0.600 |
| | YYIR1 (m) | 1,678.9 ± 207.1 | 1,734.4 ± 359.7 | 1,661.8 ± 322.5 | 0.423 | 0.657 |

**Notes.**

VO$_2$ max, maximal oxygen uptake;  SLJ,  standing long jump; 5× 25-m RSA, 5× 25-m repeated sprint ability;  YYIR1,  YoYo intermittent recovery test level 1.

sequence or structure of the VDR protein, it may modulate the stability of VDR mRNA or interfere with the transcription of the VDR gene. Consequently, this polymorphism in the VDR gene can impact the activity of the vitamin D signaling pathway by influencing the expression or function of the VDR gene, thereby affecting muscle strength (including speed and power).

Previous studies have focused on the association of *FokI* polymorphisms with muscle mass and sarcopenia in the elderly (*Bollen et al., 2023*), but the molecular and physiological basis of this association remains uncertain. Most studies have revealed that carriers of the *FokI* ff genotype performed better across a range of muscle phenotypes, and that those individuals who carried the F allele had a higher risk of sarcopenia (*Roth et al., 2004*, *Cohen et al., 2017*; *Hopkinson et al., 2008*; *Walsh et al., 2016*; *Xia et al., 2019*; *Windelinckx et al., 2007*). *Flore et al. (2024)* suggested that *FokI* led to different responses to training by increasing muscle size and strength, even in young soccer players (*Flore et al., 2024*). *Cohen et al. (2017)* suggested that males with the F allele realized greater baseline biceps strength ($p = 0.049$), and after 12 weeks of training, these males showed a smaller change

in overall muscle volume than ff homozygotes ($p = 0.034$) (*Cohen et al., 2017*). However, it has also been suggested that the FokI was not significantly associated with muscle strength or physical performance (*Bulgay et al., 2023*).

*Qi et al. (2024)* demonstrated that the *FokI* CC genotype in Chinese wrestlers was significantly higher than in their control group of ordinary college students, and that the CC genotype could be used as a molecular marker for selecting outstanding Chinese wrestlers (*Qi et al., 2024*). However, our study found that the proportion of the CC gene type in Chinese youth soccer players was not significantly different from that of the control group of ordinary male students, while the proportion of the TT gene type was significantly lower than that of the control group (see Table 4). In addition, our study showed that the distribution percentage for *BsmI* AA and AG genotypes in elite players was significantly less than that for controls and sub-elite players; therefore, the screening of *BsmI* AA and AG genotypes could be reduced in the selection process of elite athletes.

This study further elucidates the mechanisms by which VDR gene polymorphisms influence athletic performance in soccer players. From a practical standpoint, the findings of this study can provide a reference for the selection of adolescent soccer players and the development of personalized training programs, assisting coaches and training teams in better tailoring training plans according to the genetic characteristics of athletes.

There are some limitations to this study, however. Players were categorized into "elite" (starting players) and "sub-elite" (non-starting players) groups. However, this classification does not adequately account for individual differences among players, injury rehabilitation processes, or the constraint that coaches can only select one starter for players in the same position. Furthermore, for youth players, their developmental trajectories and potential may be underestimated, and this classification method may not accurately reflect players' true capabilities and future development. Our small sample size may have impacted the accuracy of the results and the values of the significant differences may also have been reduced. There is additionally a lack of distribution data on ApaI, BsmI, and FokI genotypes among Chinese professional soccer players, which is a crucial factor applied for assessing the association between genotype and athletic performance. Finally, we recommend that researchers use a longitudinal design with a larger sample size in future analyses to better assess the effects of genotype on athletic performance. We also recommend incorporating strength tests in future studies on soccer players to further elucidate the effects of the vitamin D receptor polymorphisms on muscle strength across different populations.

## CONCLUSIONS

This study found that elite Chinese youth soccer players are more likely to possess the ApaI CC genotype. Furthermore, players with the ApaI CC genotype exhibited superior explosive power and sprinting speed. Additionally, elite Chinese youth soccer players are less likely to carry the BsmI A allele. We discerned no association of ApaI, BsmI, and FokI genotypes with maximal oxygen uptake. The findings of this study may hold potential value for talent identification and individualized training among Chinese youth soccer players. Strength and conditioning coaches may thereby adjust the training load by considering

players' genetic information. In particular, Chinese youth football players carrying the ApaI CC genotype may have a greater advantage in speed and explosive power. Targeted training modalities such as short-distance sprints, plyometric box jumps, and resistance exercises may further optimize these performance attributes.

### Funding

The authors received no funding for this work.

### Competing Interests

The authors declare there are no competing interests.

### Author Contributions

- Shidong Yang conceived and designed the experiments, performed the experiments, analyzed the data, prepared figures and/or tables, authored or reviewed drafts of the article, and approved the final draft.
- Wei Zhang conceived and designed the experiments, authored or reviewed drafts of the article, and approved the final draft.
- Meng Jia performed the experiments, analyzed the data, prepared figures and/or tables, and approved the final draft.
- Haichun Chen performed the experiments, authored or reviewed drafts of the article, and approved the final draft.

### Human Ethics

The following information was supplied relating to ethical approvals (i.e., approving body and any reference numbers):

The Fujian Normal University granted Ethical approval to carry out the study within its facilities (FNU-L 2024005).

### Data Availability

The raw measurements are available in the Supplementary File.

### Supplemental Information

Supplemental information for this article can be found online at http://dx.doi.org/10.7717/peerj.19696#supplemental-information.

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
