# Peer review of "Association between vitamin D receptor gene polymorphisms and athletic performance in Chinese male youth soccer players"

_PeerJ, doi:10.7717/peerj.19696_

## Round 0.1 · original submission · Major Revisions

Based on the reviewers’ comments, your manuscript requires major revisions before further consideration. Key concerns include the unclear or retrospective nature of ethical approval, insufficient detail in describing the study design and participant characteristics (particularly regarding “elite” versus “sub-elite” athletes), and the need for more recent and relevant references in both the introduction and discussion. The reviewers also recommend providing clearer rationale for why certain performance tests were chosen, offering additional methodological specifics (e.g., consistent testing times, subject inclusion/exclusion criteria), and ensuring that your discussion aligns more closely with the data collected (e.g., focusing on tested athletic parameters rather than unmeasured factors like muscle size). Addressing these points thoroughly—along with clarifying sample selection, gene expression data presentation, and how your findings contribute novel insights—will significantly strengthen your manuscript.

·

Basic reporting

Dear Editor of the Journal
Thank you for choosing me as a reviewer
The necessity and importance of the test is not well stated
What was the sampling method based on?
This expression of this receptor is not different in strength and endurance athletes
The reasons and mechanisms should be further discussed

Experimental design

What was the sampling method based on?
This expression of this receptor is not different in strength and endurance athletes
Why is there no image of gene expression polymorphisms provided?

Validity of the findings

Why is there no image of gene expression polymorphisms provided?

Additional comments

The reasons and mechanisms should be further discussed

Reviewer 2 ·

Basic reporting

Dear authors. I congratulate you on the interesting and comprehensive article. Please find below my comments.
Title - is comprehensive and understandable
Abstract - is also comprehensive and understandable.
Article is written in a professional manner with appropriate figures and tables.

Experimental design

Introduction - is well-structured and provides a comprehensive review on the matter. Nonetheless, there are many outdated references that need to be replaced with novel ones.
The final paragraph of the introduction should be restructured in a form that describes the study's aims and provides a hypothesis (or more) to it.
Methods - I have one concern with respect to "elite" versus "sub-elite" concepts! I don't think a teen starter should be considered elite in comparison to the substitute. What if that player is recovering from a certain situation and has not managed to get the first team yet?! Or even worse, what if there are two players of the same position, out of whom the coach should decide to keep only one in the first team?! I think this should be discussed thoroughly and should be added as a major limitation in the limitations section.
Subsection 1.4 should be short and concise, whereas all the other assessments should be grouped in two subsections (e.g. anthropometrics, physical fitness assessments).

Validity of the findings

THe article provides novel insights into the matter with the potential of reproducibility.
All statistical calculations seem well conducted and understandable.
Conclusions are well stated and linked to original research.

Additional comments

Discussion - pp.235 - I would suggest to add some other (more novel) studies investigating the implication of vitamin D related genotypes and muscle phenotypes in older adults.
HWE is mentioned in the text and abstract, though I do not see anywhere in the article the outcomes from it. Please add this as well.

Reviewer 3 ·

Basic reporting

Line 16: “Previous studies have shown an association between the vitamin D receptor and muscle strength in athletes.” Why do you only mention muscle strength?

Line 29: Show more results in this section, not just p-values.

Line 37: Consider discussing the implications.

Introduction:
Focus more on athletic performance, the relationship between vitamin D and athletic performance, vitamin D in football players, and the rationale for studying vitamin D in football players.

Line 82: "One hundred and forty-two Han Chinese players."

Lines 82-86: Add inclusion and exclusion criteria, location, football experience, and subject characteristics.

Lines 85-86: Why do you define starters as elite players and substitutes as sub-elite players?

Line 88: Why was ethical approval granted in 2023, but the study was conducted in 2021/2022? Genetic sequencing and athletic performance testing of football players was completed in 2021 and the controls in 2022. Why were the football players and control subjects tested at different times? Also, who were the subjects in the control group?

Line 95: The title mentions "athletic performance," but this study only tests the 30-m sprint, standing long jump, 5×25-m shuttle run, and YYIR1. Based on the Guidelines for Testing the Athletic Ability of Youth Football Players, additional factors such as flexibility, agility, strength, and body composition should be considered.

Line 95: On the first day, testing took place from 08:00 to 10:00 AM; however, on the second day, testing occurred at the same time in the afternoon. So, what is the time for the second test?

Lines 92-98: Provide more details on the experimental methods.

Lines 242-252: You did not test strength in this study, so why do you discuss muscle strength and muscle size/mass in detail?
In the discussion section, focus more on the factors you tested, relating your findings to previous studies, not just literature from China.

Line 275: How did you reach this conclusion?

Experimental design

The research methods are unclear to the reader

Validity of the findings

The rationale and contribution to the literature are not clearly stated

Additional comments

no

---

## Round 0.2 · Major Revisions

Dear Author,

Please refer to the third reviewer's previous and current comments and address them comprehensively. Additionally, please find my comments below:

Line 20: In the abstract, what does "yet the finding of this research remain inconclusive" mean? Please use the correct phrase here.

Line 23: Use "soccer" instead of "football" to avoid confusion with American football. Please use the same term consistently throughout the manuscript.

Line 26: Please use a more appropriate term than "repeat sprint ability."

Line 73: For the sentence "Vitamin D deficiency has been linked to a number of adverse effects on muscle function, including muscle wasting, reduced muscle contraction, chronic muscle pain, and delayed recovery from muscle injury (Bulgay et al., 2023)," provide references for each association.

Line 88: Briefly justify why you conducted this research before stating the objectives.

Line 94: What does "Han" Chinese players mean?

Line 105: Please describe who collected and preserved the buccal mucosa, and how.

Line 105: Did you inadvertently omit the testing procedures?

Line 209: Begin by narrating the study's findings before describing other results.

Line 273: Clearly state the significant findings of this study. Highlight what is important, what is not, and how coaches might use this information to decide on training programs.

There are numerous grammatical and formatting errors in the manuscript, such as numbering without periods.

**Language Note:** The Academic Editor has identified that the English language must be improved. PeerJ can provide language editing services - please contact us at [email protected] for pricing (be sure to provide your manuscript number and title). Alternatively, you should make your own arrangements to improve the language quality and provide details in your response letter. – PeerJ Staff

·

Basic reporting

Article edited and its can be publish

Experimental design

Good

Validity of the findings

Good

Additional comments

File edited and its can be publish

Reviewer 3 ·

Basic reporting

The authors did not fully address my concerns and only selectively answered my questions.

For some comments, please revise the content accordingly rather than simply providing a reply.

Please cite the Youth Training Syllabus (2020 Edition) published by the Chinese Football Association in your manuscript, and explain the rationale for selecting these tests.

Ensure that all abbreviations are spelled out in full at first mention in the manuscript, e.g., standing long jump (SLJ)、Yo-Yo Intermittent Recovery Test Level 1 (YYIR1)

What level or training experience is used to define elite and sub-elite players? The reader is confused about the subject characteristics in your article, please elaborate.

Note is required under all tables

Experimental design

no comment

Validity of the findings

no comment

---

## Round 0.3 · accepted · Accept

I am happy with the changes made by the authors in the manuscript. However, few grammatical errors would be corrected during the publication process or with the help of professional scientific proofreading.